# Thermal Stability, Mechanical Properties and Ceramization Mechanism of Epoxy Resin/Kaolin/Quartz Fiber Ceramifiable Composites

**DOI:** 10.3390/polym14163372

**Published:** 2022-08-18

**Authors:** Chenyi Xue, Yan Qin, Huadong Fu, Jiamin Fan

**Affiliations:** Key Lab of Advanced Technology for Specially Functional Materials, Ministry of Education, School of Materials Science and Engineering, Wuhan University of Technology, Wuhan 430070, China

**Keywords:** epoxy resin, ceramifiable composites, mechanical properties, kaolin

## Abstract

The application of epoxy resins in high temperature and thermal protection fields is limited due to their low decomposition temperature and low carbon residual rate. In this paper, epoxy resin (EP)/quartz fiber (QF) ceramifiable composites were prepared using a prepreg-molding process. The thermal stability, phase change and mechanical properties after high-temperature static ablation and ceramization mechanism of EP/QF ceramifiable composites were investigated. The addition of glass frits and kaolinite ceramic filler dramatically increases the thermal stability of the composites, according to thermogravimetric (TG) studies. The composite has a maximum residual weight of 61.08%. The X-ray diffraction (XRD) results show that the mullite ceramic phase is generated, and a strong quartz diffraction peak appears at 1000 °C. The scanning electron microscope (SEM) and element distribution analyses reveal that the ceramic phase generated inside the material, when the temperature reaches 1000 °C, effectively fills the voids in composites. The composites have a bending strength of 175.37 MPa at room temperature and retain a maximum bending strength of 12.89 MPa after 1000 °C treatment.

## 1. Introduction

Epoxy resins are polymer materials with excellent overall performance, which are widely used in civil fields, such as coatings [1,2], electrical and electronic materials [3,4], and composite materials [5,6,7]. The demand for advanced composite materials with specific performance is growing as a result of economic and social growth. Because of their rich curing system, excellent mechanical properties and low curing shrinkage, epoxy resins are an excellent matrix for manufacturing advanced composites. Bao et al. [8] developed a flame-retardant epoxy resin using carbon nanotubes (CNTs) and silicon-containing flame-retardant (PMDA). The addition of CNTs and PMDA resulted in a 44.6% and 24.6% reduction in peak heat release rate (PHRR) and total heat release (THR) of the EP composite. Sameer A. Awad et al. [9,10] incorporated multi-walled carbon nanotube (MWCNT) into the epoxy resin to improve the resistance of the epoxy resin to UV light. Incorporating MWCNT could lead to a significantly extended useful lifetime for these materials in outdoor applications. Abdullah Al-Yami et al. [11] studied the performance of epoxy resins for application in oil and gas wells. The research indicated that EP resin–cement composite materials have excellent thermal and mechanical properties.

However, the char yield of epoxy resin is low, and it is easy to decompose into volatile organic compounds (VOCs), such as benzene, phenol, styrene, toluene, etc., and lose its strength [12], so it is seldom used in the high-temperature field above 800 °C. To solve the problem of the low carbon residue rate of epoxy resin, researchers have carried out many studies. Ling et al. [13] prepared silicone-grafted epoxy resin/carbon fiber composites. They concluded that silicone increased the carbon residue rate of epoxy resins. Compared to EP/CF composites, the mass ablation rate decreased by 43.32% at 1 MW/m^2^. Li et al. [14] prepared a polymethylphenylsiloxane-modified epoxy resin thermal protection coating, which possessed a 41% carbon residue rate at 700 °C and had excellent ablation properties. Under the heat flow of 200 kW/m^2^ for 320 s, the back-face temperature of the coating reached only 160 °C, and its surface after ablation was smooth. Existing research has focused on modifying epoxy resins by introducing polymers with high thermal stability to increase the char yield of the matrix [15,16,17]. This method has a significant effect on improving the carbon residue rate of epoxy resin, but the modification process is complicated and expensive, which is not suitable for large-scale civil applications.

In recent years, ceramifiable polymer composite materials have been widely used in the field of thermal protection system due to their excellent ablation resistance and low cost [18,19,20]. Ceramifiable polymer composites are able to maintain their strength and morphology after ablation due to the addition of high-performance ceramic fillers, which can sinter at high temperatures. When used in harsh environments, such as hypersonic aircrafts, the polymer matrices of the ceramifiable composites are often selected to have a high char yield [21]. Zou et al. [22] used boron phenolic resin, high silica fiber, and SiB_6_ to prepare the ceramifiable boron phenolic resin matrix composites. The results showed that the linear ablation rates of composites reached 7.5 µm/s, and the ceramization reaction prevented the corrosion of composites under an aerobic high-temperature environment of about 1800–1900 °C.

Researchers have added ceramic fillers to epoxy resins to obtain specific properties, such as dielectric properties [23] and wear resistance [24]. Few researchers have focused on the ceramization properties and thermal performance of EP-ceramic fillers materials. Zhao et al. introduced silicate glass frit and ammonium polyphosphate (APP) into epoxy resin to prepare a novel organic–inorganic composite [25]. The composites achieved outstanding fire resistance and flame retardance through crystallization reactions. However, few studies have been conducted to prepare fiber-reinforced ceramifiable EP-based composites, which have good mechanical properties at both room temperature and high temperatures and investigate the mechanical properties of EP-based ceramifiable composites before and after high-temperature treatment. In this paper, ceramifiable epoxy resin matrix composites were prepared by using epoxy resin as polymer matrix, quartz fiber as reinforcing material, kaolin as ceramic filler and low melting point components. The phase change and thermal stability of the ceramifiable filler system, the effects of temperature and different filler contents on the flexural properties of composites after high-temperature treatment and the high-temperature ceramization mechanism of ceramifiable epoxy resin matrix composites were studied. The ceramifiable EP/QF composites with good thermal stability and mechanical properties can be used as thermal protection materials at 1000 °C.

## 2. Experimental

### 2.1. Materials

The materials used in the experimental process were as follows: epoxy resin (EP; Araldite LY 1564; HUNTSMAN Corp., Guangzhou, China), acetone (AC; Sinopharm Group Chemical Reagent Co., Ltd., Shanghai, China), 4,4’-diaminodiphenyl sulfone (DDS; Shanghai Macklin Biochemical Co., Ltd., Shanghai, China), Kaolin (5 μm; Shanghai Macklin Biochemical Co., Ltd., Shanghai, China), low melting point glass powder (LMGP; Beginning melting temperature: 550 °C; Guangzhou Anmi Micro-nano New Materials Co., Ltd., Guangzhou, China), glass fiber powder (Softening point: 855 °C; Beginning melting temperature: 1050 °C; Chemical composition: SiO_2_: 59.3%; Al_2_O_3_: 13.5%; Fe_2_O_3_: 0.3%; TiO_2_: 0.3%; CaO: 22.4%; MgO: 2.96%; Hangzhou Corker Composites Co., Ltd., Hangzhou, China), quartz fiber mesh (SiO_2_ content > 99.9%; Hubei Feilihua Quartz Glass Co., Ltd., Jingzhou, China).

### 2.2. Preparation

#### 2.2.1. Preparation of Filler–Resin Castings

The EP, AC and curing agent DDS were mixed according to the formula in Table 1 to obtain a resin solution. Then, we mixed kaolin, glass fiber powder and LMGP, according to the formula in Table 1, and ground them with a mortar. The mixed powder was added to the mixed resin solution and stirred well. The obtained mixture was poured into the mold and cured, according to the curing process (120 °C for 1 h, 170 °C for 3 h), to obtain the filler–resin castings C1, C2, C3 and C4.

#### 2.2.2. Preparation of EP/QF Ceramifiable Composites

The EP, AC and curing agent DDS were mixed according to the formula in Table 1 to obtain a resin solution. Then, we mixed kaolin, glass fiber powder and LMGP, according to the formula in Table 1, and ground them with a mortar. The mixed powder was added to the mixed resin solution and stirred well. The obtained mixture was evenly brushed on the surface of the QF mesh to impregnate fibers thoroughly. The coated fibers were then stored at room temperature for 48 h to make them semi-dry, thus forming prepregs. The prepregs were cut into small pieces of 90 mm × 120 mm. The prepreg sheets were stacked to the specified thickness, placed neatly in the mold and cured, according to the curing process (120 °C for 1 h, 170 °C for 3 h), to obtain the composites R1, R2, R3, and R4.

#### 2.2.3. Static Ablation in a Muffle Furnace

The EP/QF ceramifiable composites were cut into samples with dimensions of 80 mm × 15 mm × 4 mm. The samples were placed in a muffle furnace at different temperatures for static ablation, and then placed in the surrounding environment for cooling to room temperature after 15 min. The ablation environment was an air atmosphere with temperatures of 400 °C, 600 °C, 800 °C, 1000 °C and 1200 °C, respectively.

### 2.3. Characterization

The bending strength was tested by universal material testing machine (model CMT6103, produced by MTS Industrial Systems, Eden Prairie, MN, USA). Standard bending specimen preparation and experimental test standards were based on GB/T 1449–2005. The loading rate was 10 mm/min.

The thermal stability of the materials was tested by an integrated thermal analyzer (NETZSCH STA449F3, Selb, Germany). Test atmosphere: air; test temperature range: room temperature to 1000 °C; heating rate: 10 °C/min.

After static ablation, the phase composition of pyrolysis products was analyzed by X-ray diffraction (XRD) analysis using a D8 ADVANCE X-ray diffractometer (Billerica, MA, USA). The samples were ground into powder and then tested in a continuous scan mode at the angle (2θ) ranging from 10 to 70 ° with a scanning rate of 5 °/min.

The samples' microscopic morphology and element distribution after ablation were recorded using a scanning electron microscope (JSM-5610LV; Japan Electronics Co., Ltd., Tokyo, Japan).

## 3. Result and Discussion

### 3.1. Analysis of Thermal Stability

The TG and DTG curves of samples C1, C2, C3, C4 and the cured epoxy resin are shown in Figure 1. The thermal decomposition rate of samples C1, C2, C3 and C4 decreases with the increase in filler content. The addition of ceramic filler significantly increases the residual weight of the material, and the increase in the filler content slows down the decomposition rate of the material.

The residual weights of the samples C1, C2, C3 and C4 at 1000 °C are 45.81%, 54.17%, 56.80% and 61.08%, respectively. The residual weight of the cured pure epoxy resin at 1000 °C is only 3.15%. The thermal decomposition parameter of C1–C4 and EP are listed in Table 2. The decomposition peak temperature of cured epoxy resin is 412.2 °C. At about 400 °C, the mass of cured epoxy resin decreases precipitously, and the TG curve of cured epoxy resin has only one decreasing step. The TG curves of the composite samples C1, C2, C3, and C4 with the addition of ceramic fillers all showed two weight loss steps. The peak thermal decomposition temperatures of the two stages were around 328 °C and 514 °C, respectively. According to other research on epoxy resin curing and filler content [26], when the filler content is high (all samples in this paper have filler content higher than 100 wt.%), the polymer monomer has limited space for movement during curing due to the steric hindrance, so the degree of curing is reduced, resulting in resin degradation at lower temperatures.

### 3.2. Analysis of the Phase Composition after Static Ablation

The XRD spectrum of the residues of sample C1 after static ablation in a muffle furnace at 400 °C, 600 °C, 800 °C, 1000 °C and 1200 °C is shown in Figure 2.

According to the traditional kaolin sintering process analysis [27], when the temperature is lower than 600 °C, the kaolinite will transform into metakaolinite by removing the hydroxyl group. At 1000 °C, the metakaolinite will precipitate SiO_2_ (quartz) to form mullite ceramics. When the temperature is further increased, quartz will convert to cristobalite.

In order to reduce the sintering temperature of the composites, low melting point glass powder (LMGP) and glass fiber powder were added. The melting temperatures of the two glass fillers form a gradient. The molten glass phase acts as a fluxing agent [28], allowing the kaolin to undergo a ceramization transformation at a lower temperature.

When the temperature is low, the primary reaction of the material is the decomposition reaction of the epoxy resin matrix. Most of the epoxy resin decomposition becomes VOCs and escapes, and a small part forms amorphous carbon inside the material. At this time, the glass fillers are not softened and melted, and the ceramic fillers are dispersed in the composite material in the form of powder. When the temperature reaches 800 °C, the cristobalite phase appears, indicating that the transformation of kaolinite to mullite has occurred [29]. When the temperature is 1000 °C, the cristobalite diffraction peak is further enhanced, and strong quartz and mullite diffraction peaks appear; at this time, the LMGP and glass fiber powder have been melted, and the quartz phase is derived from the crystallization of SiO_2_ in glass fillers and the decomposition of kaolinite.

### 3.3. Bending Strength of Ceramifiable Composites

Table 3 shows the flexural strength of R1–R4 at room temperature. With the increase in ceramic filler content, the flexural strength decreased. This is because high filler content affects the interfacial bonding of the composites.

The macroscopic morphology of EP/QF ceramifiable composites R1, R2, R3 and R4 after static ablation in a muffle furnace at a specified temperature and bending strength tests is shown in Figure 3.

The samples treated at 400 °C are black because the epoxy resin has just started to decompose, and a large amount of amorphous carbon remains in the material. The samples treated at 600 °C are white. The epoxy resin matrix decomposes at this temperature, and the molten liquid phase is not yet formed. The gas formed by the decomposition of epoxy resin escapes from the material due to the lack of protection from the ceramic phase. When the treatment temperature was 800 °C and 1000 °C, the samples are black again. The glass frits are softened and melted at this temperature, and the kaolin transforms into ceramic. The fillers form a smooth ceramic phase on the surface of the composites, which prevents further corrosion inside the material, so a certain amount of amorphous carbon is retained [30]. At 1200 °C, the epoxy resin matrix decomposes violently, and VOCs escape before the ceramic phase is formed, so the samples are whiter in color than the samples treated at 1000 °C.

Figure 4 shows the change of bending strength of R1–R4 after static ablation at different temperatures.

The bending strengths of R1, R2, R3 and R4 are 40.57 MPa, 39.22 MPa, 38.70 MPa and 35.79 MPa, respectively, when the samples were statically ablated in a muffle furnace at 400 °C for 15 min. The epoxy resin just reaches the decomposition temperature, and a large number of resin decomposition products reside in the material so that the material still maintains a high bending strength. When the material is treated at a lower temperature, the epoxy resin matrix still maintains its strength, while the ceramization process has not yet started. So, the higher content of ceramic fillers decreases the material’s strength.

When the treatment temperature is 600 °C, the bending strength of the samples decreases significantly. The epoxy resin matrix is cracked as VOCs, while only the LMGP has just reached the beginning melting temperature. The composite is mainly composed of quartz fibers, filler powder and amorphous carbon, and there is a lack of connection between these phases, so the bending strength of the material is significantly reduced.

The bending strength of R1 and R2 after treatment at 800 °C and 1000 °C are lower than that of the samples treated at 600 °C, while R3 and R4 are on the contrary. This is because when the temperature reaches 800 °C or higher, the resin matrix loses the ability to bond to the fiber. The glass filler has reached the temperature of softening and melting, forming a liquid phase to fill the material voids and act as a binder in the material. The result is that the more fillers there are, the tighter the connection between the composites is.

The micromorphology of the interior of samples R1–R4 after ablation at 1000 °C is shown in Figure 5.

There are fewer residues inside R1 and R2, the ceramic phase does not coat the fibers well, and there are large voids between the fibers. The internal residues of R3 and R4 are wrapped around the fibers, the gaps between fibers are well filled, and a large area of molten liquid phase appears in R4.

When the composite material is under load, the fibers play the role of bearing the load in the composite material, while the matrix plays the role of bonding and transferring the load. As shown in Figure 6, when the epoxy resin decomposes, there are a large number of voids in the material, and there is no continuous phase between the fibers to transfer the load. The stress leads to the destruction of the matrix phase, which has a poor load-bearing capacity, so the strength of the composite is low. When there is enough ceramics filler in the composite, although the epoxy resin decomposes and produces a large number of voids, the glass filler melts and liquefies, filling the voids between the fibers quickly [31]. When the material is under load, the stress is transferred from the charred matrix and ceramic phases to the fibers, so the composite still has some strength.

The bending strength of the sample after static ablation at 1200 °C is significantly reduced because the quartz fibers are fused and stuck together, resulting in strength failure (shown in Figure 7). In addition, the viscosity of the glass filler is low at 1200 °C, coupled with the violent decomposition of the epoxy resin at this temperature, which makes the interior of the composite washed out by the gas, and a large number of voids appear again.

The stress–strain curves of sample R4 after different temperature treatments are also shown in Figure 4. The results clearly show that R4 shows a significant decrease in bending modulus at 600 °C, which is consistent with the conclusion obtained above: the material matrix is mainly composed of loose material. The modulus of the sample rises again when the temperature is above 800 °C, which is due to the ceramization reaction and the gradual transformation of the polymer material to a ceramic material. It is worth noting that brittle fractures occurred in the samples treated at 1000 °C and 1200 °C.

### 3.4. Analysis of the Ceramization Mechanism

Figure 8 shows the micromorphology and element distribution of sample R4 after static ablation at different temperatures. 

At a treatment temperature of 400 °C, the fibers are tightly bound to each other and pores of small diameters appear between the matrix. From the element distribution analysis in Figure 8c, it is clear that the main composition of the matrix is the charring product of epoxy resin. The carbon content decreases dramatically when the processing temperature is 600 °C, and many gaps are created between the fibers, due to the decomposition of the epoxy resin and the formation of VOCs. In addition to C, the matrix is mainly composed of O, Si and Al, which are SiO_2_ and Al_2_O_3_ in kaolin and glass frits.

When the treatment temperature is 800 °C, the content of the elements in the matrix remains basically the same as at 600 °C. When the treatment temperature is 1000 °C, the material is tightly combined internally, and a large area of liquid phase appears. From the element distribution analysis in Figure 8k,l, it can be seen that the composition of the liquid phase has a high content of Si and a shallow content of Al. Combined with the XRD analysis above, the material shows a firm quartz diffraction peak at 1000 °C. Therefore, the liquid phase is mainly precipitated by the crystallization of SiO_2_ in glass frits and kaolinite.

At room temperature, the ceramic filler is uniformly dispersed in the resin matrix, as shown in Figure 9a. When the temperature reaches the decomposition temperature of the epoxy resin (about 400 °C), the ceramic filler is still dispersed in the matrix in the form of powder, as shown in Figure 9b. The epoxy resin decomposes into amorphous carbon at this time, but it has not yet escaped from the composite and still maintains the bonding ability. When the temperature continues to rise to 600 °C, the epoxy resin matrix starts to release gases (VOCs) violently. At this time, the LMGP has melted to form the liquid phase, but the glass fiber powder has not yet softened, and the kaolin has not undergone the ceramization transformation. Therefore, as shown in Figure 9c, the internal structure of the composite is not well bonded. When the temperature reaches about 1000 °C, as shown in Figure 9d, the glass phase acts as a fluxing agent to lower the ceramization temperature of kaolinite, and the ceramic phase generated inside the material fills the voids created by the decomposition of the epoxy resin and binds the fiber bundles together.

## 4. Conclusions

In this paper, EP/QF ceramifiable composites were successfully prepared. The epoxy resin-based composites with added ceramic fillers have good static ablation properties, thermal stability and mechanical properties after high-temperature treatment. The addition of the ceramic fillers significantly increases the residual weight of the material, and the elevated filler content slows down the material’s decomposition rate. The glass fillers reduce the ceramization transition temperature of kaolinite to 800 °C.

The shape and size of the EP/QF ceramifiable composites after static ablation at 400–1200 °C in a muffle furnace are basically stable, and 12.89 MPa bending strength is retained after 15 min static ablation in a muffle furnace at 1000 °C. The ceramic phase generated inside the material when the temperature reaches 1000 °C effectively fills the voids created by the decomposition of the epoxy resin, binding the fiber bundles together and solving the problem of the low carbon residual rate of the epoxy resin.

## Figures and Tables

**Figure 1 polymers-14-03372-f001:**
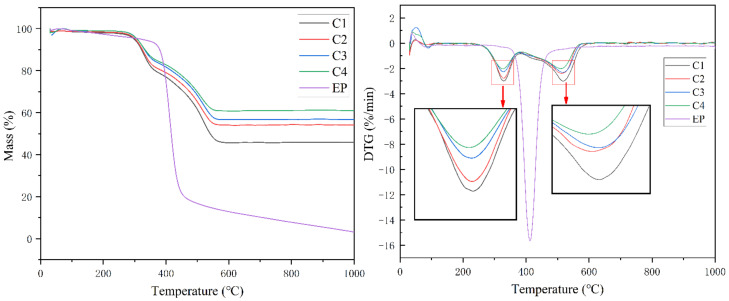
TG and DTG curves of samples C1, C2, C3, C4 and EP.

**Figure 2 polymers-14-03372-f002:**
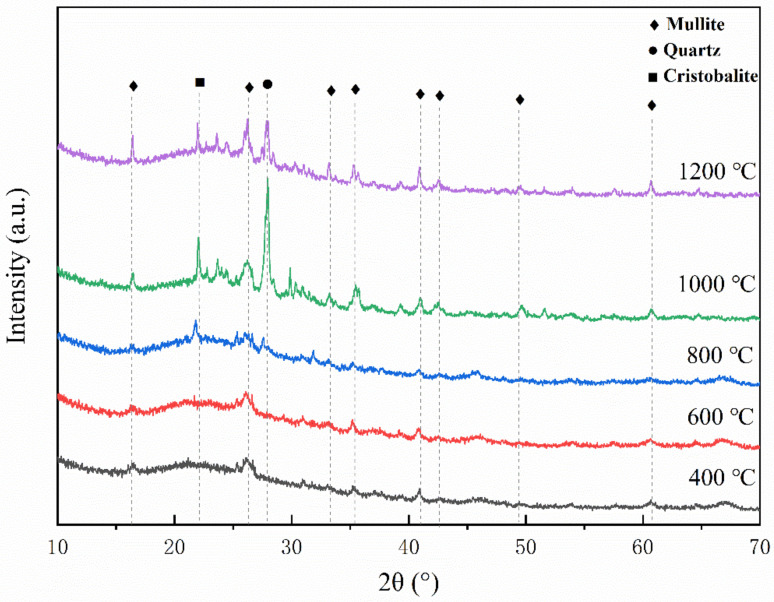
XRD spectrum of sample C1 after static ablation at 400 °C, 600 °C, 800 °C, 1000 °C and 1200 °C.

**Figure 3 polymers-14-03372-f003:**
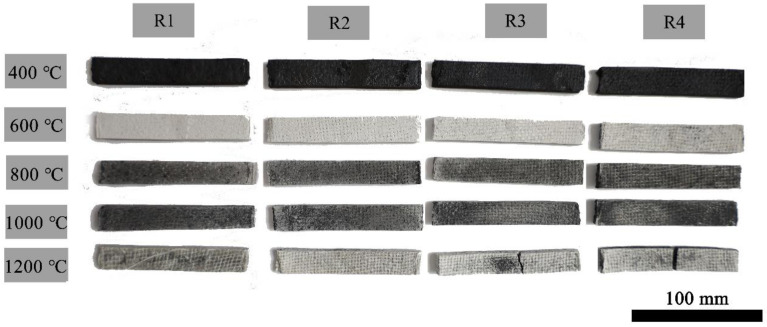
Macroscopic morphology of EP/QF ceramifiable composites after static ablation and blending strength tests.

**Figure 4 polymers-14-03372-f004:**
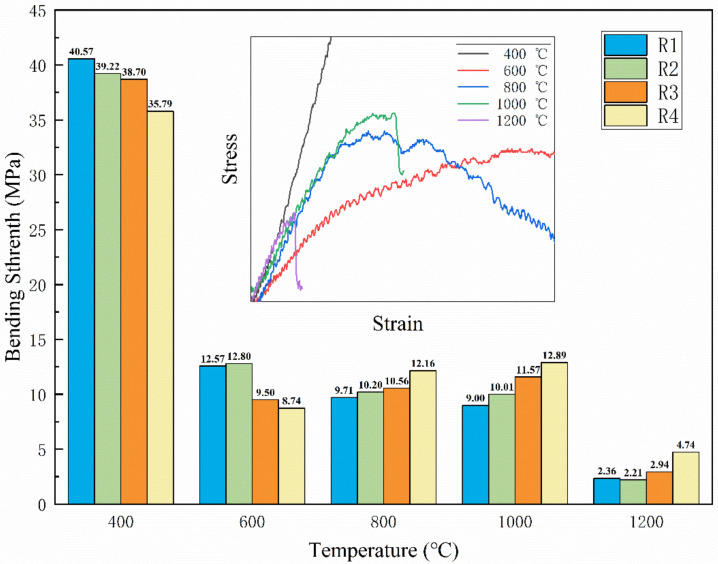
Bending strength of R1–R4 after static ablation at different temperatures.

**Figure 5 polymers-14-03372-f005:**
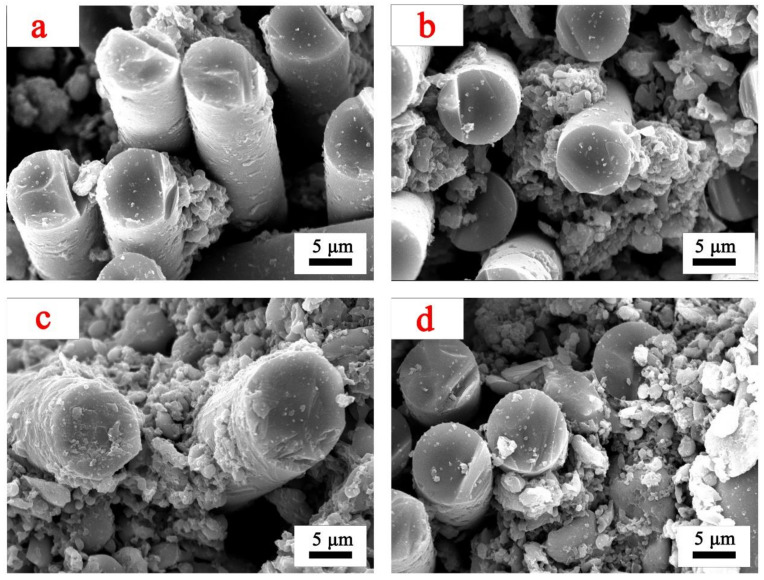
Micromorphology of samples treated at 1000 °C: (**a**) micromorphology of sample R1, (**b**) micromorphology of sample R2, (**c**) micromorphology of sample R3, (**d**) micromorphology of sample R4.

**Figure 6 polymers-14-03372-f006:**
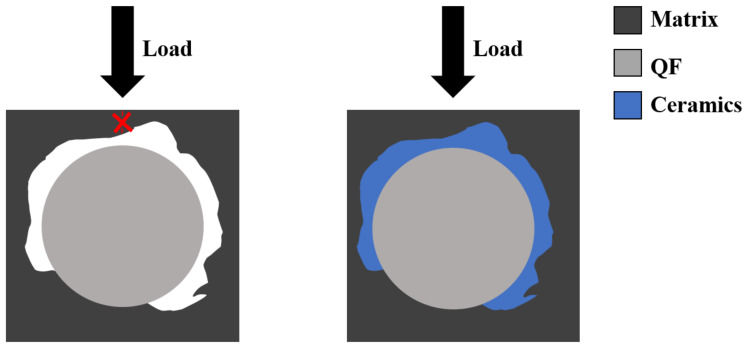
Failure mechanism of EP/QF ceramifiable composites.

**Figure 7 polymers-14-03372-f007:**
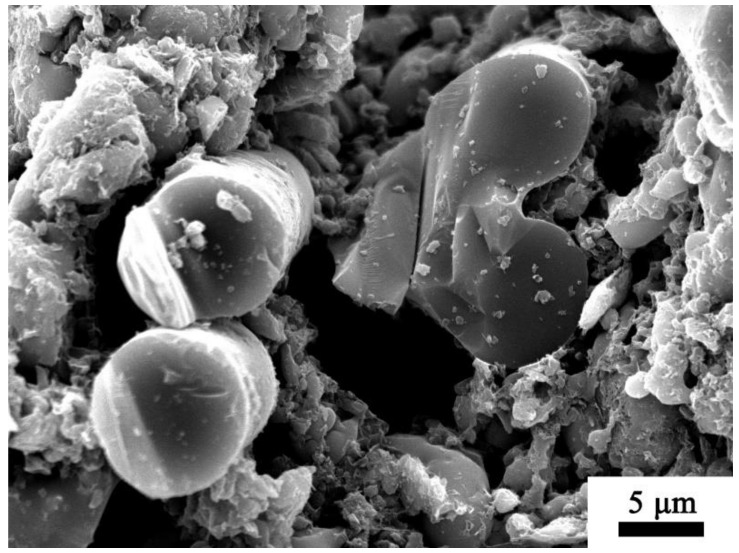
Micromorphology of sample R4 treated at 1200 °C.

**Figure 8 polymers-14-03372-f008:**
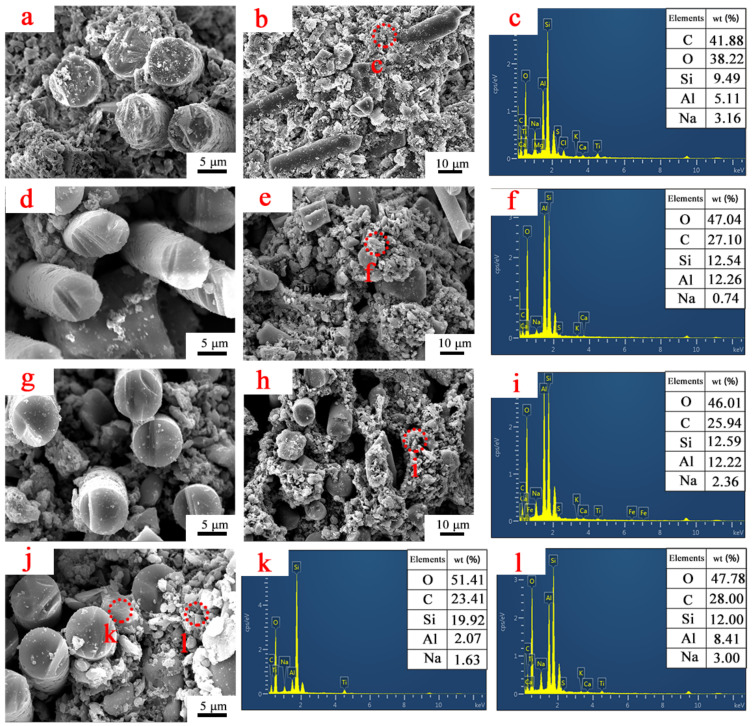
The micromorphology and element distribution of sample R4 after static ablation at different temperatures: (**a**,**b**) micromorphology of R4 after static ablation at 400 °C, (**c**) element distribution; (**d**,**e**) micromorphology of R4 after static ablation at 600 °C, (**f**) element distribution; (**g**,**h**) micromorphology of R4 after static ablation at 800 °C, (**i**) element distribution; (**j**) micromorphology of R4 after static ablation at 1000 °C, (**k**,**l**) element distribution.

**Figure 9 polymers-14-03372-f009:**
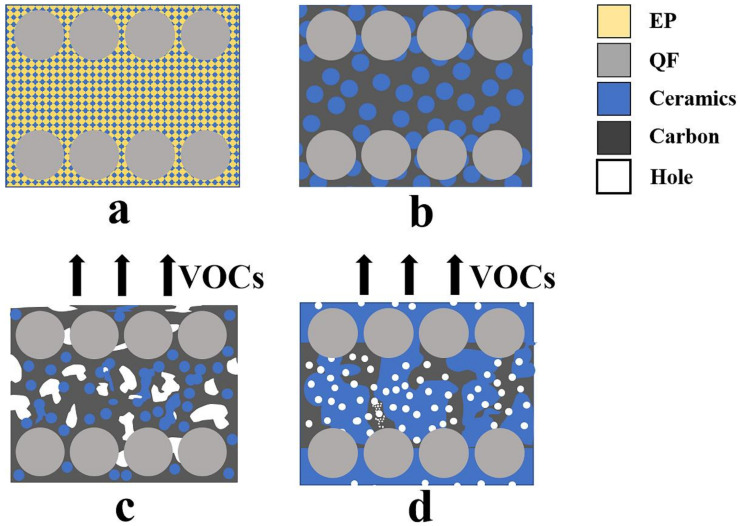
Ceramization mechanism of EP/QF ceramifiable composites.

**Table 1 polymers-14-03372-t001:** The formula of the filler–resin castings and EP/QF ceramifiable composites.

Sample	EP (phr)	AC (phr)	DDS (phr)	Kaolinite (phr)	GF Powder (phr)	LMGP (phr)
C1/R1	100	50	30	55	27.5	27.5
C2/R2	100	50	30	70	35	35
C3/R3	100	50	30	85	42.5	42.5
C4/R4	100	50	30	100	50	50

**Table 2 polymers-14-03372-t002:** Thermal decomposition parameter of C1–C4 and EP.

Sample	Tmax1 (°C)	Tmax2 (°C)	Residue Rate (%)
C1	329.8	518.0	45.81
C2	328.9	513.0	54.17
C3	328.8	517.7	56.80
C4	326.8	510.3	61.08
EP	412.2	–	3.15

**Table 3 polymers-14-03372-t003:** Bending strength of R1–R4 at room temperature.

Sample	Bending Strength (MPa)
R1	268.29
R2	223.05
R3	201.86
R4	175.37

## Data Availability

The datasets generated during and/or analyzed during the current study are available from the corresponding author on reasonable request.

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
