# Peer review of "Thermal Stability, Mechanical Properties and Ceramization Mechanism of Epoxy Resin/Kaolin/Quartz Fiber Ceramifiable Composites"

_polymers, 2022, doi:10.3390/polym14163372_

Round 1

Reviewer 1 Report

Xue et al. reports the sintering process of epoxy resin-based inorganic/organic composites. The system is simple: epoxy, hardener, ceramifiable Kaolinite, glass fiber and glass powder, all mixed, and cured. Then, the authors investigated the sintering behaviors of the cured composites at different temperature. Based on the observations of TGA, XRD, bending strength, and morphologies, the authors claimed that the ceramifization mechanisms are revealed.

However, the current manuscript fails to show complete picture. Also the authors should elaborate many aspects in Introduction and discussions. I feel that the authors should improve the manuscript to be publishable in Polymers. Below are the points:

1. In Introduction, the authors stated that “There are few researches on the use of epoxy resin as a matrix in the field of ceramifiable composites.”, however, there are no description on the reports. The authors should elaborate what has been done so far with epoxy-based ceramifiable composite, and what has to be addressed by the authors’ work.

2. In TGA (Figure 1), it is clear that the composite has facilitated the degradation of epoxy resin compared to only EP. What is it? Why it starts to degrade at lower temperature? The authors should rationalize this observation.

3. The authors stated that “Most of the epoxy resin decomposition becomes VOCs and escapes”, what types of VOCs are generated during epoxy decomposition?

4. “The epoxy resin just reaches the decomposition temperature, a large number of resin decomposition products residue in the material so that the material still maintains a high bending strength, which is also the reason why the less ceramic filler content, the higher the bending strength.” What does this mean? To rationalize this hypothesis, the authors should examine the bending strength of the composite before sintering. 

5. Figure 5: It does not seem that the fibers in Figure 5d are well-coated by residues. In Figure 7, also the fibers are not well-coated by residues. Figure 8d also shows the fiber not well-coated by resin, which is treated even at the lowest temperature. The mechanism in Figure 6 is accordingly not well understood. The authors would need more careful examination on the morphology.

6.  In Figure 2, 1000 C sample exhibits very strong peak at ~28 degree. This goes weaker when the temperature increased to 1200 C. Why does this occur? Is quartz formed and deformed? 

7. Title: “QF” should not be abbreviated, should be changed to full name.

8. Conclusion should be written in one or two paragraphs. Currently, one sentence is one paragraph.

9. Reference formatting is required.

Author Response

Response to Reviewer 1 Comments

Thank you very much for your suggestions on my article. I have revised my article after reading your suggestions carefully and added some experiments. My responses to the comments you made are as follows:

Point 1: In Introduction, the authors stated that “There are few researches on the use of epoxy resin as a matrix in the field of ceramifiable composites.”, however, there are no description on the reports. The authors should elaborate what has been done so far with epoxy-based ceramifiable composite, and what has to be addressed by the authors’ work.

Response 1: Researchers have added ceramic fillers to epoxy resins to obtain specific properties such as dielectric properties and wear resistance. Few researchers have focused on the ceramization properties and thermal performance of EP-ceramic fillers materials. Zhao et al. introduced silicate glass frit and ammonium polyphosphate (APP) into epoxy resin to prepare a novel organic-Inorganic composite. The composites achieved outstanding fire resistance and flame retardance through crystallization reactions. However, few studies have been conducted to prepare fiber-reinforced ceramifiable EP-based composites which has good mechanical properties at both room temperature and high temperatures and investigate the mechanical properties of EP based ceramifiable composites before and after high temperature treatment.

Point 2: In TGA (Figure 1), it is clear that the composite has facilitated the degradation of epoxy resin compared to only EP. What is it? Why it starts to degrade at lower temperature? The authors should rationalize this observation.

Response 2: According to other researches on epoxy resin curing and filler content (Zhao, Y.; Drummer, D. Influence of Filler Content and Filler Size on the Curing Kinetics of an Epoxy Resin. Polymers 2019,11,1797.), when the filler content is high (all samples in this paper have filler content higher than 100 wt.%), the polymer monomer has limited space for movement during curing due to the steric hindrance, so the degree of curing is reduced, resulting in resin degradation at lower temperatures.

Point 3: The authors stated that “Most of the epoxy resin decomposition becomes VOCs and escapes”, what types of VOCs are generated during epoxy decomposition?

Response 3: According to the GC-MS test in our previous work, the products of resin decomposition at high temperature include benzene, phenol, styrene, toluene and other organic compounds containing benzene rings. Since the types of decomposition products have no effect on the ceramicization process of composite materials, the GC-MS test results are not included in this paper. According to your question, I have listed the main types of VOCs in the introduction.

Point 4: “The epoxy resin just reaches the decomposition temperature, a large number of resin decomposition products residue in the material so that the material still maintains a high bending strength, which is also the reason why the less ceramic filler content, the higher the bending strength.” What does this mean? To rationalize this hypothesis, the authors should examine the bending strength of the composite before sintering.

Response 4: This conclusion means: When the material is treated at a lower temperature, the epoxy resin matrix still maintains its strength, while the ceramization process has not yet started. So, the higher content of ceramic fillers makes the strength of the material decrease instead. Results for flexural strength at room temperature have been obtained which consistent with the above conclusion. To make the conclusion more understandable, I have improved the expression.

Point 5: Figure 5: It does not seem that the fibers in Figure 5d are well-coated by residues. In Figure 7, also the fibers are not well-coated by residues. Figure 8d also shows the fiber not well-coated by resin, which is treated even at the lowest temperature. The mechanism in Figure 6 is accordingly not well understood. The authors would need more careful examination on the morphology.

Response 5: In epoxy resin based ceramifiable composites, the degradation of epoxy resin is inevitable. The high content of ceramic filler does not perfectly form a matrix without voids, but it can be observed from the mechanical properties and SEM that it effectively binds the fibers.

Although the fibers in Fig. 5d is not completely wrapped by the matrix, there is clearly more residue between the fibers than in Fig. 5 a,b,c. And there is liquid phase with larger area in the matrix in Fig. 5d.

Figure 7 shows an image of the material after treatment at 1200°C. the viscosity of the glass filler is low at 1200℃, coupled with the violent decomposition of the epoxy resin at this temperature, which makes the interior of the composite washed out by the gas, therefore a large number of voids appear again.

I have modified Figure 6 to make it easier to understand. There is no continuous phase between the fibers to transfer the load. The stress leads to the destruction of the matrix phase which has a poor load-bearing capacity, so the strength of the composite is low. With the addition of more ceramic fillers, the stress is transferred from the charred matrix and ceramic phases to the fibers, so the composite still has some strength.

Point 6: In Figure 2, 1000 C sample exhibits very strong peak at ~28 degree. This goes weaker when the temperature increased to 1200 C. Why does this occur? Is quartz formed and deformed?

Response 6: According to the article “Evolution of Mullite Texture on Firing Tape-Cast Kaolin Bodies” (has been added to References), when the temperature is lower than 600℃, the kaolinite will transform into metakaolinite by removing the hydroxyl group. At 1000℃, the metakaolinite will precipitate SiO2 (quartz) to form mullite ceramics. When the temperature is further increased, quartz will convert to cristobalite.

The above explanation has been added to the article.

Point 7: Title: “QF” should not be abbreviated, should be changed to full name.

Response 7: The title has been changed into “Thermal stability, mechanical properties and ceramization mechanism of Epoxy Resin/Kaolin/Quartz Fiber ceramifiable composites”.

Point 8: Conclusion should be written in one or two paragraphs. Currently, one sentence is one paragraph.

Response 8: The conclusion has been changed into two paragraphs.

Point 9: Reference formatting is required.

Response 9: The format of the references has been modified.

In addition, with the help of my colleagues who are good at English, I have improved the English expression of the article to make it more readable.

Reviewer 2 Report

Comments to authors are listed below:

  • The abstract lacks to present the numerical values from significant findings in this paper.
  • The last paragraph of the introduction should be reported with the novelty and applications of this work.
  • The introduction should be improved by including the specific properties and applications of epoxy resins. Some references are below, should be included:

https://www.ncbi.nlm.nih.gov/pmc/articles/PMC8513003/

https://www.sciencedirect.com/science/article/pii/S0142941817313065

https://journals.sagepub.com/doi/abs/10.1177/0021998318816784

https://link.springer.com/article/10.1007/s13369-018-3651-y

  • Tensile stress-strain tests should be performed to show improve the quality of the mechanical tests section.
  • Discussion of the results is not enough and should be improved by interpreting and clarify the significant presents in detail and should compare with previous work to show the difference in improvements.
  •  

Author Response

Response to Reviewer 2 Comments

    Thank you very much for your suggestions on my article. I have revised my article after reading your suggestions carefully and added some experiments. My responses to the comments you made are as follows:

Point 1: The abstract lacks to present the numerical values from significant findings in this paper.

Response 1: I have added the findings of the mechanical properties, the thermal stability analysis and XRD analysis to the abstract with numerical values.

Point 2: The last paragraph of the introduction should be reported with the novelty and applications of this work.

Response 2: The novelty and applications of this work has been added in the last paragraph of the introduction. Researchers have added ceramic fillers to epoxy resins to obtain specific properties such as dielectric properties and wear resistance. Few researchers have focused on the ceramization properties and thermal performance of EP-ceramic fillers materials. Zhao et al. introduced silicate glass frit and ammonium polyphosphate (APP) into epoxy resin to prepare a novel organic-Inorganic composite. The composites achieved outstanding fire resistance and flame retardance through crystallization reactions. However, few studies have been conducted to prepare fiber-reinforced ceramifiable EP-based composites which has good mechanical properties at both room temperature and high temperatures and investigate the mechanical properties of EP based ceramifiable composites before and after high temperature treatment.

Point 3: The introduction should be improved by including the specific properties and applications of epoxy resins. Some references are below, should be included:

Response 3: I have carefully studied the four pieces of literature you provided. They describe special properties and applications of epoxy resins (flame retardant, UV resistance, application in oil well environment, etc.), which are very suitable to add to the introduction part of my article about epoxy resins in advanced materials. I have added them to the introduction section.

Point 4: Tensile stress-strain tests should be performed to show improve the quality of the mechanical tests section.

Response 4: The ceramifiable composites after high temperature treatment have one thing in common: low compressive strength. When performing tensile tests, the specimen should be held in place with a fixture according to the tensile test standard. In practice, the pressure generated by the fixture can cause damage to the material. Therefore, bending strength is generally used as a standard when testing the mechanical properties of ablated ceramifiable composites. The following studies are examples:

1.DOI 10.3390/polym14101944

2.https://doi.org/10.1002/fam.3059

3.https://doi.org/10.1080/14658011.2020.1731258

I have added the flexural stress-strain curve of R4 in Figure 4, which can also illustrate the ceramicization process of the material and more intuitive.

From the results, it is clear that R4 shows a significant decrease in bending modulus at 600°C, which is consistent with the conclusion obtained above: the material matrix is mainly composed of loose material. The modulus of the sample rises again when the temperature is increased above 800°C, which is due to the ceramization reaction and the gradual transformation of the polymer material to a ceramic material. It is worth noting that brittle fracture occurred in the samples treated at 1000°C and 1200°C.

Point 5: Discussion of the results is not enough and should be improved by interpreting and clarify the significant presents in detail and should compare with previous work to show the difference in improvements.

Response 5: According to your comments I have revised the discussion section, mainly including the following aspects:

1) I added a clarification to the TG section, mainly adding an explanation of the thermal decomposition rate.

2) I added an explanation about the elevated and then lowered quartz diffraction peaks in the XRD analysis section.

3) In the mechanical properties section, I added experiments related to the bending strength of the material at room temperature and the stress-strain curve after ablation. This allows a more comprehensive description of the mechanical properties of the material and the process of ceramization.

According to our investigation, no studies on fiber-reinforced epoxy resin-based ceramifiable composites were found. Therefore, the study on the porcelain-forming mechanism and mechanical properties of fiber-reinforced composites in this paper provides a reference for other researchers and provides an idea for the application of epoxy resin matrix composites in high temperature field.

In addition, with the help of my colleagues who are good at English, I have improved the English expression of the article to make it more readable.

Round 2

Reviewer 1 Report

.

Reviewer 2 Report

No comments.